# Bioprotection Efficiency of *Metschnikowia* Strains in Synthetic Must: Comparative Study and Metabolomic Investigation of the Mechanisms Involved

**DOI:** 10.3390/foods12213927

**Published:** 2023-10-26

**Authors:** Maëlys Puyo, Perrine Mas, Chloé Roullier-Gall, Rémy Romanet, Manon Lebleux, Géraldine Klein, Hervé Alexandre, Raphaëlle Tourdot-Maréchal

**Affiliations:** 1UMR Procédés Alimentaires et Microbiologiques, Institut Agro Dijon, Université de Bourgogne Franche-Comté, 21000 Dijon, France; maelys.puyo@gmail.com (M.P.); chloe.roullier-gall@u-bourgogne.fr (C.R.-G.); lebleuxmanon@gmail.com (M.L.); geraldine.klein@u-bourgogne.fr (G.K.); rvalex@u-bourgogne.fr (H.A.); 2DIVVA (Développement Innovation Vigne Vin Aliments) Platform/PAM UMR A 02.102, IUVV, 2 Rue Claude Ladrey, 21000 Dijon, France; remy.romanet@sayens.fr

**Keywords:** bioprotection, enology, nitrogen resource, *Metschnikowia*, metabolomic analyses, oxygen consumption

## Abstract

Three *Metschnikowia* strains marketed as bioprotection yeasts were studied to compare their antimicrobial effect on a mixture of two *Hanseniaspora* yeast strains in synthetic must at 12 °C, mimicking pre-fermentative maceration by combining different approaches. The growth of the different strains was monitored, their nitrogen and oxygen requirements were characterised, and their metabolomic footprint in single and co-cultures studied. Only the *M. fructicola* strain and one *M. pulcherrima* strains colonised the must and induced the rapid decline of *Hanseniaspora*. The efficiency of these two strains followed different inhibition kinetics. Furthermore, the initial ratio between *Metschnikowia* and *Hanseniaspora* was an important factor to ensure optimal bioprotection. Nutrient consumption kinetics showed that apiculate yeasts competed with *Metschnikowia* strains for nutrient accessibility. However, this competition did not explain the observed bioprotective effect, because of the considerable nitrogen content remaining on the single and co-cultures. The antagonistic effect of *Metschnikowia* on *Hanseniaspora* probably implied another form of amensalism. For the first time, metabolomic analyses of the interaction in a bioprotection context were performed after the pre-fermentative maceration step. A specific footprint of the interaction was observed, showing the strong impact of the interaction on the metabolic modulation of the yeasts, especially on the nitrogen and vitamin pathways.

## 1. Introduction

As an alternative to the antimicrobial properties of sulphites, bioprotection can be defined as the inoculation of selected microorganisms directly on grape during harvest or early on must, in order to colonise the environment during the pre-fermentative steps and thus limit the impact of indigenous flora. Many microorganisms, yeasts and bacteria, are naturally present on the grape berry and found in must after pressing [1,2]. Among them, a large majority can survive for only a short time in the must due to the stressful conditions induced by alcoholic fermentation. However, the apiculate yeasts of the *Hanseniaspora* genus, representing almost 80% of native yeasts, as well as *Brettanomyces bruxellensis*, may persist during fermentation and produce undesirable compounds such as acetic acid or 4-ethyl phenol [3,4,5,6], which subsequently strongly impact the organoleptic quality of wines.

New starters are already marketed as bioprotective yeasts [7]. These starters are mainly non-*Saccharomyces* (NS) yeasts. Among them, strains belonging to the *Metschnikowia* genus are those most frequently recommended. In red winemaking, an *M. pulcherrima* bioprotective strain was used on Pinot Noir by adding rehydrated active dry yeast to grapes during vatting. Its implementation during pre-fermentative steps at 12 °C successfully limited the growth of the spoilage flora [8]. The same strain was tested in white winemaking on Chardonnay at different pre-fermentative racking temperatures (7, 12, and 18 °C). The strain was successfully implanted in all conditions except at 18 °C, with a significant effect on the initial native population decline (greater than 1 log) [9]. The bioprotective effect of another *M. pulcherrima* strain was also confirmed during the cold clarification of a Verdicchio must [10]. The use of *M. pulcherrima* in combination with *Torulaspora delbrueckii* also showed that mixing these two NS yeasts was able to limit the risk of spoilage microorganism growth when applied early in the winemaking process [11,12]. However, Windholtz et al. (2021) highlighted that the implantation and thus the effectiveness of the *M. pulcherrima*/*T. delbrueckii* mixture was weaker under advanced grape maturity conditions [13].

Although the positive effect of selected strains of *Metschnikowia* has already been demonstrated in cellar conditions, few laboratory experiments have been conducted to elucidate negative interactions occurring on grape must between a bioprotective strain and indigenous flora. In the oenological context, numerous interactions between yeasts, such as the production of toxic compounds like toxins killer, or nutrient competition have already been suggested [14,15,16,17,18]. This study compares the efficiency of three *Metschnikowia* strains (two *M. pulcherrima* strains and one *M. fructicola* strain) in artificially contaminated synthetic must at 12 °C to fit with the pre-fermentative maceration conditions applied in the cellar. Two levels of contamination were tested, mimicking native yeast population rates that may fluctuate depending on grape ripening conditions. In view of investigating the physiological mechanisms implied in the effectiveness of the bioprotection, we characterised the nitrogen and oxygen needs of the strains and analysed the metabolomic footprint of the different growth conditions.

## 2. Materials and Methods

### 2.1. Yeast Strains

Three commercial yeasts provided by the French institute of vine and wine (Institut Français de la Vigne et du Vin (IFV), Nantes, France) were studied as bioprotectants: two *Metschnikowia pulcherrima* strains coded Mp1 and Mp2 and one *Metschnikowia fructicola* strain coded Mf1. To mimic the indigenous flora, a 50%/50% mix of *Hanseniaspora uvarum* (3137) and *Hanseniaspora valbyensis* (ScS) strains isolated in the laboratory from grape berries was used.

Bioprotective yeasts were stored in active dry yeast (ADY) forms. *Hanseniaspora* strains were stored in glycerol (Sigma-Aldrich, Saint-Quantin-Fallvier, France) 20% at −80 °C.

### 2.2. Culture Conditions and Sampling

Before use, ADYs were rehydrated following the manufacturer’s instructions. Regarding *Hanseniaspora*, each strain was plated on a petri dish in solid YPD medium (Glucose 20 g/L, yeast extract 5 g/L, tryptone 10 g/L, chloramphenicol 0.2 g/L, and agar 16 g/L) and incubated at 28 °C for 48 h. A single colony was collected and pre-cultured in liquid YPD at 28 °C for 48 h under agitation (130 rpm). The pellet of each strain (*Metschnikowia* and *Hanseniaspora*) obtained after centrifuging 1 mL of pre-culture or rehydration (5 min at 5000 rpm) was resuspended in MacIlvain buffer (Na_2_HPO_4_ 28.39 g/L, citric acid (anhydrous) 19.21 g/L). For each sample, 20 µL of cells in MacIlvain buffer was then analysed through flow cytometry (BD Accuri^TM^ C6 Plus, Becton, Dickinson and Company, Franklin Lakes, NJ, USA) at a flow rate of 34 µL/min, with an FSH threshold of 80,000. To obtain the total cell concentration, 1 µL of carboxy-fluorescein diacetate (cFDA) (Invitrogen, Thermo Fisher Scientific, Waltham, Massachusetts, USA) at an initial concentration of 1500 µmol/µL was added to 100 µL of culture in MacIlvain buffer to obtain the concentration of viable cells, using an argon laser at 488 nm for cell excitation (cell autofluorescence) and a 533/530 nm band pass filter to measure cFDA fluorescence. The viable cell concentration obtained through flow cytometry of the pre-culture of rehydrated cells was used to inoculate the culture flask.

Cultures were conducted in triplicate in 100 mL sterile Erlenmeyer flasks, closed with carded cotton, containing 80 mL of sterile synthetic must MS300 (adapted from Bely et al. (1990) [19] and described in Evers et al. (2023) [20]). For the monocultures, *Metschnikowia* yeasts were inoculated at an initial concentration of 1.10^6^ viable cells/mL (initial concentration recommended by the manufacturers) and *Hanseniaspora* yeasts were inoculated in a 50/50 mixture (*H. uvarum*/*H. valbyensis*) at 5.10^4^ and 5.10^6^ viable cells/mL to simulate two levels of contamination (low contamination and high contamination, as observed on some grapes at advanced maturity) (Table 1). Each bioprotectant strain was also inoculated (1.10^6^ cells/mL) in co-culture with the *Hanseniaspora* mixture at both alteration levels 5.10^4^ and 5.10^6^ cells/mL (Table 1). Cultures were maintained for 48 h at 12 °C (time/temperature pairing often used in the pre-fermentation phase). For each flask, 500 µL of culture (mono- and co-cultures) was sampled sterilely at 0 h, 8 h, 24 h, 32 h, and 48 h after the inoculation of the bioprotectant yeast and *Hanseniaspora* mixture; 100 µL was used for cell enumeration and the remaining 400 µL was stored at −20 °C for chemical analyses.

All the chemical products used for medium were purchased from Sigma-Aldrich (Saint-Quantin-Fallvier, France).

### 2.3. Growth Monitoring

For each sampling, populations of *Metschnikowia* strains were enumerated by counting on WL Oxoïd CM0309 (Oxoïd, Hampshire, UK) medium with the addition of 0.2 g/L of chloramphenicol (Sigma-Aldrich, Saint-Quantin-Fallvier, France), in which they produce red colonies due to pulcherriminic acid production [21] after 48 h incubation at 28 °C. Populations of *Hanseniaspora* were counted on modified ITV selective medium (20 g/L glucose, 10 g/L yeast extract, 20 g/L tryptone, 0.1 g/L para-coumaric acid, 0.1 g/L ferulic acid, 0.03 g/L green bromocresol, 0.2 g/L chloramphenicol, and 20 g/L agar, pH 5, with the addition of cycloheximide 0.006% (*v*/*v*) [22] after 48 h incubation at 28 °C.

### 2.4. Nitrogen Compound Analyses

#### 2.4.1. Ammonia and Total Amino Acid Analyses

Ammonia and total amino acid concentrations of samples were analysed using an automated Y15 analyser (BioSystem, Muttenz, Switzerland) with the “Primary Amino Nitrogen (PAN)” kit, which is a colorimetric analyser, and the “Ammonia” kit, which provides enzymatic analysis (BioSystem, Muttenz, Switzerland).

#### 2.4.2. Amino Acid Analyses

Amino acid concentrations were measured through high-pressure liquid chromatography (HPLC) with an HPLC Agilent 1260 Infinity (Agilent, Santa Carla, CA, USA), using a Poroshell 120 (100 × 4.6 mm) column with a particle size of 2.6 µm (Agilent, Santa Carla, CA, USA). The analytical method used was the Agilent amino acids method with automated o-phtaldehyde (OPA) (Agilent, Santa Carla, CA, USA)/9-fluorenylmethylchloroformate (FMOC) (Agilent, Santa Carla, CA, USA) derivatisation, according to the Agilent application note number 5991-5571EN (https://www.agilent.com/cs/library/applications/5991-5571EN.pdf, accessed on November 2, 2017). The analytical method used a mobile phase A composed of Na_2_HPO_4_ (1.4 g/L) (Sigma-Aldrich, Saint-Quantin-Fallvier, France) and 10 mM Na_2_B_4_O_7_.10H_2_O (3.8 g/L) (Sigma-Aldrich, Saint-Quantin-Fallvier, France), adjusted to pH 8.20, and a mobile phase B composed of a mixture of acetonitrile/methanol/water (45/45/10, *v*/*v*/*v*). A constant flow rate was set up to 2 mL/min, using the gradient presented in Appendix A, and a column oven at 40 °C. Before analyses, samples were centrifuged at 10,500× *g* for 10 min. Amino acid auto-derivatisation was conducted as follows: 2.50 µL of borate buffer (placed in a 2 mL vial in the autosampler) was collected in the sample loop with 0.50 µL of sample and mixed twice in the loop. After waiting for 30 s, 0.50 µL of OPA (placed in a 2 mL vial in the autosampler) was collected and added in the loop and mixed 6 times with the mixture of borate and sample. After waiting for 18 s, 0.50 µL of FMOC (placed in a 2 mL vial in the autosampler) was collected and mixed 6 times. After waiting for 24 s, 32 µL of injection eluant (eluant A + 4 mL/L H_3_PO_4_, placed in a 2 mL vial in the autosampler) was added in the loop and mixed with the mixture of OPA/FMOC/sample; then, 0.5 µL of the derivatised sample was injected in the column.

### 2.5. Oxygen Analyses

Measurements of yeast oxygen consumption were performed in synthetic must MS300. Cultures were performed in triplicates at 12 °C in glass bottles of 100 mL containing 100 mL of MS300, previously saturated in O_2_, and inoculated with strains (monoculture), as for growth monitoring (Table 1). Oxygen concentration was determined with the Nomasens^®^ P300 (Vinventions WQS ©, Rivesaltes, France) analyser with Pst3 pads (Vinventions WQS ©, Rivesaltes, France), which were glued inside the glass bottle, until all the dissolved oxygen was consumed.

### 2.6. Untargeted Analysis Using UHPLC-q-TOF-MS/MS

Non-volatile compounds were analysed for the three biological replicates of each condition (sampled at 48 h of growth) through ultra-high-pressure liquid chromatography (UHPLC) (Dionex Ultimate 3000, ThermoFisher, Waltham, MA, USA) coupled with a MaXis plus MQ ESI-q-TOF mass spectrometer (MS) (Bruker, Bremen, Germany), as used by Evers et al. (2023) [20].

Samples were centrifugated at 10,500× *g* for 10 min and kept at 10 °C during analysis. The nonpolar compounds were separated with reverse-phase liquid chromatography on an UPLC Acquity BEH C18 1.7 µm column 100 × 2.1 mm (Waters, Guyancourt, France), using eluant A (5% (*v*/*v*) acetonitrile with 0.1% (*v*/*v*) formic acid) and eluant B (acetonitrile with 0.1% (*v*/*v*) formic acid). The analysis used a flow rate of 0.4 mL/min and the following gradient: 5% (*v*/*v*) solvent B from 0 to 1.10 min, before increasing its proportion from 1.10 to 6.40 min to reach 95%, until the end of the analysis (10 min in all). Sample ionisation was performed in both negative and positive modes, with the nebuliser pressure set at 2 bar and a dry nitrogen flow set at 10 L/min. The mass spectrometer was set as follows: 500 V as endplate offset for ion transfer, capillary voltage at 4500 V (positive ionisation mode) and 3500 V (negative ionisation mode), mass range acquisition between 100 and 1000 *m*/*z*, and fragmentation at 8 Hz spectra rate using the auto MS/MS function (20–50 eV). External calibration was performed with the injection of Na formate before sample analysis in enhanced quadratic mode (with error < 0.5 ppm), and with the regular injection of quality control (sample mixture) to check the analysis repeatability.

Bruker Compass MetaboScape Software (v. 8.0.1, Bruker, Mannheim, Germany) was used for mass pre-treatment (mass spectrum recalibration, the extraction of *m*/*z* and the retention time of each feature, and chemical annotation (elemental formula with carbon, hydrogen, oxygen, nitrogen and sulphur atoms) with the integrated tools SmartFormula using an isotopic profile (tolerance at mSigma < 20, and 5 ppm)). The features extracted corresponded to those with an intensity higher than 1000, which were present in 20% of total replicates and 40% of recursive parameters. The features selected through statistical analysis were assigned putative chemical families (lipids, peptides, aromatic compounds, carbohydrates, and amino sugars) based on elemental formulas according to the compound classification of Rivas-Ubach et al. [23] and hypothetical annotation using inhouse database as well as the online databases Yeast Metabolome Database (YMDB) and Kyoto Encyclopedia of Genes and Genomes (KEGG). The annotation level was established according to the description in the study of Schymanski et al. [24].

### 2.7. Statistical Analyses

The analysis of growth parameters, graphics production, and statistical analyses were performed with RStudio Software (version 4.2.2, Boston, MA, USA). Growth and nitrogen and oxygen consumption parameters were statistically analysed using the Kruskal–Wallis test and Dunn’s post hoc test with an α error of 5%. For the metabolomic analyses, the relative intensities of features were analysed using the *t*-test (*n* = 2) and ANOVA followed by a Tukey post hoc test when needed (*n* = 3) with an α error of 5%.

## 3. Results and Discussion

### 3.1. Growth Parameters of Metschnikowia Strains in Monocultures

The growth kinetics of each *Metschnikowia* strain were first characterised in single cultures. At 12 °C, which corresponds to the temperature commonly used for pre-fermentative maceration, the three *Metschnikowia* strains showed similar growth rates (µmax) (Table 2), but differences were observed regarding the lag phase time. The lag time was 32 h before the growth of the Mp1 strain, compared with 8 h for the Mf1 and Mp2 strains. The Mp1 strain also exhibited the lowest growth, with only a 4-fold increase in biomass after 48 h compared to its initial cell concentration, contrary to strains Mp2 (14-fold increase) and Mf1 (11-fold increase) (Table 2). The growth differences observed for strain Mp1 could be explained by the longer lag phase time compared to the other two strains, which suggested a longer temperature adaptation phase for this strain.

### 3.2. Bioprotectant Efficiency

For co-cultures, two contamination levels with the *Hanseniaspora* mix were used (5.10^4^ and 5.10^6^ CFU/mL) to reproduce two contamination levels like those on grapes with low and advanced maturity. The inoculation rate of each bioprotective strain was 1.10^6^ CFU/mL. The growth curves are presented in Figure 1. To determine the effect of the interactions on the growth of each bioprotection/apiculate yeast pair, kinetics parameters were compared in single and co-culture for each initial concentration of the *Hanseniaspora* mix. The growth parameters of the single cultures were considered control values to estimate changes resulting from the interaction (Table 3).

For the *M. pulcherrima* Mp1 strain inoculated with 5.10^4^ CFU/mL of *Hanseniaspora*, its growth parameters were similar to those found in single culture (Table 3). When inoculated with 5.10^6^ CFU/mL of *Hanseniaspora*, growth was negatively impacted by the interaction (*p* < 0.05), with a slowdown in the specific growth rate value and less than a 2-fold increase in the initial population (Table 3).

Regarding the growth of *Hanseniaspora* when inoculated with strain Mp1, whatever the initial *Hanseniaspora* concentration, the µmax values of the apiculate yeasts were not affected by the interaction with strain Mp1 (*p* > 0.05) (Table 4, Figure 1a). The final population of *Hanseniaspora* was statistically lower in co-culture, but this difference was not biologically relevant, with a difference of less than ½ log (Table 4, Figure 1). We also observed a slight growth slowdown after 32 h in co-culture with an initial concentration of 5.10^4^ CFU/mL (Figure 1b).

In our conditions, strain Mp1 exhibited weak growth, which was also negatively impacted with the highest initial *Hanseniaspora* concentration. Furthermore, this strain was not able to inhibit or significantly slow down the growth of *Hanseniaspora* whatever the initial concentration of apiculate yeast. According to the low impact of the *M. pulcherrima* Mp1 strain on the implantation and persistence of *Hanseniaspora* in the medium, we can conclude that the bioprotective effect of the Mp1 strain in our condition was negligible.

Regarding the co-cultures of Mp2 or Mf1 strains with *Hanseniaspora* at an initial concentration of 5.10^4^ CFU/mL, the *M. pulcherrima* Mp2 strain was negatively impacted by the interaction with a significantly slower specific growth rate value and a lower final population (Table 3, Figure 1b). However, the statistical difference for the final population was not biologically relevant (2.37.10^7^ (±0.23.10^7^) CFU/mL and 1.40.10^7^ (±0.41.10^7^) CFU/mL for single and co-cultures, respectively). The *M. fructicola* Mf1 strain was not impacted by the interaction regarding its growth rates (Table 3), but it reached a lower final population (*p* < 0.05). In this condition, both *Metschnikowia* strains were negatively impacted by the interaction but in different ways; the growth rate of Mp2 was impacted while the final population of Mf1 was changed (Table 3, Figure 1b,c).

Regarding their bioprotective effect with an initial population of 5.10^4^ CFU/mL of *Hanseniaspora*, both strains showed significant changes in the growth parameters of the *Hanseniaspora* mix. When inoculated with strain Mp2, *Hanseniaspora* was unable to grow. The population persisted for 24 h before declining (Table 4, Figure 1b). The *Hanseniaspora* population in co-culture with the Mf1 strain exhibited a similar µmax value to that in single culture, but a lower final population (*p* < 0.05) (Table 4, Figure 1c). In co-culture with Mf1, the *Hanseniaspora* population exhibited similar growth kinetics to those in single culture for the first 24 h, but interaction with Mf1 led to its death after 24 h of growth (Figure 1c).

When *Hanseniaspora* was inoculated at an initial concentration of 5.10^6^ CFU/mL with Mp2 or Mf1 strains, the final *Hanseniaspora* populations were significantly lower, with a reduction of approximatively 1 log (Table 4, Figure 1b,c). Only the co-culture with the Mf1 strain induced a weak effect on the specific growth rate value of *Hanseniaspora* (*p* ≈ 0.049) (Table 4, Figure 1c). Regarding the implantation of *Metschnikowia* yeasts, Mp2 reached a similar final concentration as in single culture and faster growth but exhibited a lag phase of 32 h, while in single culture, its growth started quickly. The Mf1 strain reached a significantly lower final population in co-culture (single culture: 1.18.10^7^ (±0.25.10^7^) CFU/mL, co-culture: 2.60.10^6^ (±0.51.10^6^) CFU/mL) (Table 3, Figure 1c). In this condition, although its final concentration was lower in co-culture than in single culture, *Hanseniaspora* population was not greatly inhibited by the *Metschnikowia* strains, which did not exhibit any significant antagonist effect.

The three *Metschnikowia* strains tested showed different profiles. *M. pulcherrima* Mp1 was the only strain that showed limited implantation at 12 °C linked to the absence of a bioprotective effect. These results suggest that this strain was unsuited for our experimental conditions, whereas field trials had previously demonstrated the colonisation of grape must by bioprotective strains of *M. pulcherrima* during cold pre-fermentative maceration [9,10], as well as its ability to grow at 12 °C in YNB synthetic medium [25]. Albertin et al. [25] have also reported low-temperature tolerance concerning *H. uvarum* strains at 12 °C in laboratory conditions.

With an initial *Hanseniaspora* concentration of 5.10^4^ CFU/mL, strains Mp2 and Mf1 were characterised by a strong bioprotective effect, but their efficiency was different. This result suggests a potentially different path of action for antagonist effects, whether strain- or species-dependent. However, the hypothesis of the existence in both strains of similar mechanisms such as the production of antimicrobial compounds, cell–cell contact, or competition for other nutrients, but with different kinetics, cannot be discarded. This inhibitory effect was strongly limited in the presence of a large population of *Hanseniaspora*. The initial ratio between the bioprotective yeast and *Hanseniaspora* must be taken into account to optimise bioprotection efficiency in cellar conditions. During field trials, the lack of bioprotection established on grapes at an advanced stage of ripening was highlighted previously, with the initial high concentration of indigenous yeasts as the main hypothesis [13].

### 3.3. Nitrogen Consumption

To explain the antagonistic effect observed previously, an analysis of nitrogen compounds consumed by each yeast was carried out to investigate whether these effects were due to a potential competition for these resources.

The analysis of the nitrogen resource consumption by the Mp1 strain revealed a very low consumption of available nitrogen (<1% of total amino acids) (Table 5), linked to weak growth at 12 °C. Therefore, more detailed analyses were focused on strains Mp2 and Mf1. Neither strain consumed the amino acids in their entirety after 48 h growth in single culture. Lysine was the amino acid most consumed, with 60% (Mp2) and 51% (Mf1) consumed after 48 h growth and only slight ammonia consumption (≤10%) (Table 5). Nitrogen consumption for *Hanseniaspora* was higher for the highest inoculation rate (24% of the total amino acids consumed in the highest concentration and only 6% in the lowest one) (Table 5, Appendix A). However, whatever the initial concentration of *Hanseniaspora*, there was no total consumption of nitrogen resources. For the initial concentration of 5.10^4^ CFU/mL, lysine was the only amino acid to be consumed at more than 50% (66% consumed), contrary to the initial concentration of 5.10^6^ CFU/mL where aspartic acid, isoleucine, leucine, lysine, methionine, and threonine were consumed at more than half of their initial concentrations in 48 h (Table 5).

To further investigate a potential competition for nitrogen resources between bioprotective strains and *Hanseniaspora*, we chose to focus on the common amino acids consumed most under all conditions: isoleucine, leucine, lysine, methionine, and tryptophan (Table 5).

Lysine was the amino acid consumed most by both *Metschnikowia* strains, followed by leucine (30%), tryptophan, and isoleucine (20% for both strains). The consumption of tryptophan started rapidly (after 7 h and 3 h of culture for Mp2 and Mf1 strains, respectively) (Figure 2, Appendix A). However, this high initial consumption stopped after 30 h of culture. It is also worth noting that isoleucine consumption kinetics differed between the two strains. Strain Mf1 started to consume isoleucine after 8 h growth, whereas the Mp2 strain consumed it only after 24 h. But the same amount was consumed by both strains in 48 h.

Due to protocol variability (strain, medium, temperature, agitation, initial nitrogen concentration, etc.), the comparison of the data with the literature is complex. However, lysine, leucine, and isoleucine were also referred to as “good”/“preferred” or “intermediate” nitrogen resources for *M. pulcherrima* in other studies conducted on grape juice or synthetic media (YNB or synthetic must) [26,27,28]. In our conditions, both the *Metschnikowia* species tested had similar nitrogen consumptions. These results are supported by Su et al. [29], who also found that *M. fructicola* and *M. pulcherrima* strains had comparable nitrogen preferences. Furthermore, Su et al. [29] showed that in synthetic must with an initial nitrogen concentration of 300 mgN/L, the main three amino acids consumed by *Metschnikowia* strains are lysine, isoleucine, and leucine [29]. Regarding methionine and tryptophan, which were considered to be consumed in our results, the literature is more ambivalent. Some studies reported tryptophan or methionine as being weakly consumed and others as being consumed by the *Metschnikowia* strain [26,27,28,29].

For these five amino acids and for the initial concentration of 5.10^6^ CFU/mL in *Hanseniaspora*, their consumptions were highest with a low lag phase, compared with the results obtained for the lowest inoculation rate. Indeed, among the selected amino acids, only tryptophan was consumed at less than 50% after 48 h, and lysine was almost entirely consumed (95%). Lysine was also the amino acid most consumed in the condition with an initial *Hanseniaspora* concentration at 5.10^4^ CFU/mL (66% consumed), while the other amino acids selected were consumed at less than 30% (Figure 2, Table 5). In the literature, very little information is reported about the nitrogen consumption of *H. uvarum* or *H. valbyensis*. According to the data available, the five selected amino acids consumed by *H. uvarum* are also reported to be “good” or “intermediate” resources [26,28]. Among the other resources consumed at more than 10%, aspartic acid was also consumed by *Hanseniaspora*, especially for the initial condition of 5.10^6^ CFU/mL, as well as by the Mp2 strain (Table 5, Appendix A). However, the Mp2 strain consumed this resource only over the first 8 h, before stopping to leave about 70% of the initial aspartic acid concentration in the medium. Aspartic acid consumption was also reported in the literature for *H. uvarum* [26,28].

The resources preferentially consumed by *Metschnikowia* were also found to be preferentially consumed by *Hanseniaspora*. Although the consumption rate of *Hanseniaspora* at 5.10^6^ CFU/mL was significantly slower (in ng/L/h/cell) in this condition for isoleucine and tryptophan (Appendix A), their percentages in terms of consumption were higher due to the high cell concentration.

Regarding co-cultures with an initial 5.10^4^ CFU/mL of *Hanseniaspora* in which a bioprotective effect was observed for the Mp2 and Mf1 strains, both the *Metschnikowia* strains presented very similar consumption profiles. However, the timing of the antagonistic effect differed between the strains. Indeed, Mp2 inhibited the growth of *Hanseniaspora* before causing its decline, while Mf1 allowed *Hanseniaspora* to grow for the first 24 h before also causing its death. Since these two strains had comparable yeasts assimilable nitrogen (YAN) consumption profiles, if the bioprotective effect was due to competition for an amino acid essential for their growth, then the bioprotective effect of the two *Metschnikowia* strains would be similar. Moreover, the consumption for competing resources was only partial in single cultures and in co-culture (a maximum of 30% of the total amino acids consumed, and no ammonia consumption); thus, nitrogen was not limiting in our conditions, and therefore could not lead to the death of *Hanseniaspora* (Appendix A).

Nitrogen competition cannot explain such rapid *Hanseniaspora* cell death in co-culture with Mp2 or Mf1 in our conditions.

### 3.4. Oxygen Requirement

Oxygen consumption at 12 °C was measured over time. For the three bioprotective strains, the oxygen in the medium was completely consumed between 2 and 3 h, with a significantly higher consumption rate for the Mp1 strain (Figure 3, Appendix A). The same result was obtained with *Hanseniaspora* at a concentration of 5.10^6^ CFU/mL. The slowest consumption was obtained with an initial population of 5.10^4^ CFU/mL of *Hanseniaspora* with total oxygen depletion in the medium after about 34 h (Figure 3, Appendix A). Little research has been reported about the oxygen requirement of NS yeasts in must. In co-inoculation with *S. cerevisiae*, several NS yeasts including *M. pulcherrima* exhibited a better survival time during fermentation with the addition of oxygen [30,31,32,33,34]. These results seemed to indicate the importance of oxygen in the medium and its contribution to yeast growth and persistence. In the literature, *Metschnikowia pulcherrima* was reported to be a strong oxygen consumer, with a higher dependency on oxygen for its viability during fermentation with *S. cerevisiae* [32]. Regarding *H. uvarum* and *H. valbyensis,* Visser et al. (1990) showed their ability to grow anaerobically, but with an impact on their growth with a longer lag phase before growth initiation [35].

The results obtained do not seem to confirm the hypothesis of competition for oxygen to explain the antagonistic effects observed previously. Indeed, the two effective strains Mp2 and Mf1 showed the same consumption kinetics, while their inhibitory action on *Hanseniaspora* differed. Furthermore, monocultures of *Hanseniaspora* showed growth at 12 °C despite the depletion of the medium in oxygen within a few hours. The competition for this resource could explain a slowdown in their growth but not the death of the *Hanseniaspora* in the environment.

### 3.5. Untargeted Metabolomic Analyses

Untargeted metabolomic analyses allowed screening, without prior expectations, a large number of compounds to compare single and co-cultures. These analyses were designed to potentially detect metabolomic pathways or specific compounds implicated in the inhibition of the *Hanseniaspora* mix, and so they elucidate the mechanisms involved in bioprotection.

Each triplicate was analysed using UHPLC-q-TOF-MS/MS at the final time point (48 h of growth at 12 °C) for single and co-cultures with each pair of the *Metschnikowia*/*Hanseniaspora* mix.

On the principal component analysis (PCA), each triplicate was represented according to the first two dimensions (14.3% for dimension 1 and 8.7% for dimension 2) (Figure 4). The samples were separated based on the relative intensity of the 3028 features detected during the analysis.

According to this representation, several groups of samples were found. Firstly, all the triplicates were grouped together, demonstrating good reproducibility of the experiments and analyses. The bioprotective strains in single cultures were discriminated according to species by the two dimensions. *Hanseniaspora* single cultures and co-cultures were separated by the initial *Hanseniaspora* concentration, and co-cultures appeared closer to the single cultures of *Hanseniaspora* than the *Metschnikowia* single culture according to the second dimension. All the co-cultures with an initial concentration of *Hanseniaspora* at 5.10^6^ CFU/mL displayed a metabolomic footprint similar to that of the *Hanseniaspora* (5.10^6^ CFU/mL initial) single cultures. Regarding the co-cultures with 5.10^4^ CFU/mL of *Hanseniaspora*, co-cultures with Mp1 were comparable to the *Hanseniaspora* single cultures, unlike co-cultures with Mf1 and Mp2. This separation was consistent with the previous observations on growth kinetics with an initial *Hanseniaspora* concentration of 5.10^4^ CFU/mL. Indeed, co-cultures with Mp1 were found to be very similar metabolically to *Hanseniaspora* single cultures, explained by the lack of bioprotective effect of the strain. Furthermore, the Mp1 strain exhibited little growth in this condition, contrary to *Hanseniaspora*, which predominated in co-culture. In contrast, co-cultures with Mf1 and Mp2 were more distant from the *Hanseniaspora* single culture with a stronger difference in metabolomic footprint for the co-culture with Mp2. The difference in inhibition kinetics between these two strains may explain the effect observed in the metabolic footprint associated with each of the two strains tested in co-culture. To investigate the nature of these different metabolic footprints in greater detail, we chose to focus on only single cultures of the three strains of *Metschnikowia* and the *Hanseniaspora* single and co-cultures at an initial concentration of 5.10^4^ CFU/mL for subsequent data analyses.

By comparing the three bioprotective strains, 2840 features were extracted from the UHPLC-q-TOF-MS/MS analyses. Among them, only 9, 7, and 10 features were unique to Mp1, Mp2, and Mf1 strains, respectively (Appendix A). Other features were common to at least two strains. The two *M. pulcherrima* strains shared the highest number of features (107 in common), while Mp1/Mf1 had 59 common features, and 72 between Mf1 and Mp2. Among all the features extracted, 2576 of them were present in the three conditions.

Statistical analysis revealed the characteristic features of each strain. These features, called biomarkers, correspond to features unique to one condition or common to several but significantly more intense (*p* < 0.05) in the condition observed. In total, 72, 79, and 66 biomarkers were found for Mp1, Mp2, and Mf1, respectively. However, differences in chemical composition were observed between strains. Among the biomarkers of the three strains, CHON and CHONS compounds were strongly represented, like CHO compounds, except for the Mp1 condition, which had a clear predominance of CHON compounds and less CHO-type compounds (Appendix A). The association of these biomarkers with putative chemical families showed a high proportion of peptides, notably for Mp1, but differences were observed among the other potential families with a stronger presence of lipids for Mp2 and aromatic compounds for Mf1.

Among the 2576 features common to the three strains, 1796 were found to exhibit similar intensity (non-statistically different *p* > 0.05) in the three single cultures. These features represent the metabolomic footprint of the *Metschnikowia* genera, whatever the strain considered in this study. As for the biomarkers of the single cultures, the common ones seem to be mainly CHON and CHONS compounds, associated mainly with lipids (30%) and aromatic compounds (32%), as well as peptides (23%) (Appendix A).

For each *Metschnikowia*/*Hanseniaspora* (5.10^4^ CFU/mL initial population) pair, the biomarkers of each group (single cultures of *Metschnikowia* and *Hanseniaspora* and co-cultures) were extracted through comparative analysis of the co-culture with the two related single cultures.

For the Mp1/*Hanseniaspora*/Mp1+*Hanseniaspora* mix group, 2890 features were extracted, with 299 considered biomarkers for the Mp1 single culture and 43 for the *Hanseniaspora* single culture. The largest number of biomarkers obtained with the monoculture of Mp1 indicated that the metabolism of this strain was the most impacted by the interaction. In co-culture, 30 biomarkers were obtained, 13 of them unique, probably resulting from the interaction between Mp1 and *Hanseniaspora* (Appendix A). For all conditions, the biomarkers seemed to be mainly associated with CHON compounds, but differences were observed between the proportion of the other elemental compositions with a large proportion of CHONS compounds for the Mp1 single-culture and co-culture biomarkers. The classification of the biomarkers in putative chemical families showed a large proportion of peptides (linked with the high proportion of CHON and CHONS compounds possibly corresponding to peptides and sulphur-containing peptides) for Mp1 and co-culture conditions, while the *Hanseniaspora* biomarkers seemed to be equally associated to aromatic compounds, lipids, and peptides (between 26 and 30%) (Appendix A).

The same reasoning for the data analyses was applied to the other two bioprotective strains. Comparative analyses of the 2880 features extracted between Mp2 (125 biomarkers), *Hanseniaspora* (45 biomarkers), and the associated co-culture (37 biomarkers) showed a large number of possible CHON- and CHONS-type compounds among their biomarkers (Appendix A). The Mp2 single culture was the one with the most possible CHONS compounds. Lipids seemed to be equally represented among the biomarkers of all three conditions (29–35%), while peptide was the chemical family represented most for the biomarkers of Mp2 and co-culture. As for Mp1, the Mp2 single culture presented the highest number of biomarkers, whose relative quantity seemed to be negatively impacted by the interaction (Appendix A). In contrast to the previous comparison with the Mp1 strain, the chemical composition of the co-culture biomarkers shared more similitude with *Metschnikowia* biomarkers than with those of *Hanseniaspora*.

Regarding the last group with Mf1, *Hanseniaspora*, and their co-culture, 2742 features were extracted from the analysis. Interestingly, there were no features detected as unique to the *Hanseniaspora* single culture. The statistical comparison resulted in 186, 50, and 29 biomarkers for Mf1, *Hanseniaspora*, and the co-culture, respectively (Appendix A). The biomarkers of Mf1 were represented mainly by CHONS compounds, unlike *Hanseniaspora* and the co-culture, which also had a large proportion of CHON compounds associated with a high number of putative peptide compounds for the co-culture condition.

Biomarkers of the three co-cultures were compared to investigate the bioprotective mechanisms (Appendix A and Figure 5). The vast majority of these biomarkers were unique to each co-culture, showing the specific impact of the interaction with each of the *Metschnikowia* strains tested (Appendix A). Indeed, out of the 90 biomarkers extracted, whatever the co-culture, no biomarker was common to the three co-cultures. Only a few biomarkers were common between at least two co-cultures, with two and four biomarkers common between the co-cultures with Mp1 and Mf1, and co-cultures with Mp1 and Mp2, respectively (Appendix A).

Among the co-culture biomarkers, CHON-type compounds were strongly represented in the three conditions, but co-culture with Mp2 led to a higher proportion of CHONS compounds. Among the four elementary classes, co-cultures with Mp1 and Mf1 shared more similarities, but regarding the putative chemical families, co-cultures with Mp2 and Mf1 (the two conditions with a bioprotective effect) showed a higher proportion of compounds associated with potential peptides and lipids, unlike co-culture with Mp1, which was represented mainly by peptides and aromatic compounds (Figure 5).

The KEGG and YMDB databases, as well as an in-lab database, were used to annotate the biomarkers of the three co-cultures compared previously.

Among the unique biomarkers of co-culture with Mp1, nine were putatively annotated. One of them seemed to correspond to tryptophol (annotation level 2). It was interesting to note that tryptophol could be produced from tryptophan [36], which is one of the amino acids most consumed by *Hanseniaspora*. In the literature, tryptophol is described as a quorum-sensing molecule (QSM) in *C. albicans* and *S. cerevisiae* [36,37,38], although for the latter, its role is still controversial. The production of tryptophol was already reported in *H. uvarum* [39], but further investigations are needed to determine its implication as a QSM.

Regarding co-culture with Mp2, 13 biomarkers were putatively annotated by comparison with databases. Fumaric acid (annotation level 4) was one of them. This compound was reported as an intermediate of the TCA cycle (Krebs cycle). TCA cycle needs oxygen to be fully performed and completed as a cycle. In *S. cerevisiae* under anaerobic condition, as is the case in our conditions, the TCA cycle no longer works as a cycle but follows two different pathways (a reductive and an oxidative pathway) [40]. The presence of fumaric acid as a biomarker could suggest that the interaction between Mp2 and *Hanseniaspora* induced changes in the regulation of the respiro-fermentative pathways. It has already been shown in the literature that the respiro-fermentative metabolism of *S. cerevisiae* is affected by interaction with *M. pulcherrima*. Indeed, Sadoudi et al. [41] showed an impact of the interaction at the genetic level, with a significant impact of *M. pulcherrima* on the pyruvate dehydrogenase bypass and glyceropyruvic fermentation pathways regulation of *S. cerevisiae* [41]. Another biomarker corresponded to pantothenic acid (annotation level 4), a vitamin which is essential for yeast growth and its metabolism, and implicated in fatty acid production [42]. The interaction between Mp2 and the *Hanseniaspora* led to a higher pantothenic acid concentration in medium, linked either to lower consumption of this compound by yeasts in interaction, production by cells, or release in medium during *Hanseniaspora* cell death. We also noticed that tryptophan (annotation level 1) was observed among the Mp2 co-culture biomarkers. Tryptophan was among the most consumed amino acids. Its annotation as a biomarker could suggest an impact of its interaction with amino acid metabolisms.

Eight of the biomarkers extracted from co-culture with Mf1 were annotated. Annotation results suggested the presence of dethiobiotin (annotation level 4), an intermediate of biotin synthesis [43]. The interaction could involve the inhibition of the final step of biotin synthesis, which converts dethiobiotin into biotin. As for the co-culture with the Mp2 strain, the regulation of vitamin metabolism appeared to be impacted by the interaction between Mf1 and *Hanseniaspora*. Biotin plays a key role in fatty acid and amino acid metabolism [20,44], which could suggest an impact on nitrogen and fatty acid pathway regulation through biotin changes in the context of interaction. An impact on branched amino acid metabolism was also suggested by the putative annotation of another biomarker such as valine (annotation level 4) in co-culture with Mf1.

Furthermore, among the common biomarkers between the two co-cultures, we assigned a putative annotation for two of them, glutamine (annotation level 4) and 3-isopropylmalate (annotation level 4). Those two compounds were involved in amino acid metabolism. Actually, glutamine has a key role in amino acid biosynthesis, and 3-isopropylmalate was found in the literature as involved in leucin biosynthesis in yeasts [45,46], which is one of the preferred amino acids consumed for all strains in our conditions. A third compound was annotated as 4-hydroxy-L-threonin (annotation level 4). This compound was found in the KEEG pathway as involved in B_6_ vitamin metabolism (KEEG ID C06056, map00750).

Previous studies have already shown that interaction between yeasts leads to specific metabolomic footprints, with an impact not only at the level of the species analysed, but also at the level of the strain [47,48,49]. Nitrogen and vitamin metabolism appeared to be impacted by the interaction between *Hanseniaspora* and *Metschnikowia* strains and could participate in the bioprotective effect. Further research on the nitrogen and vitamin metabolism changes in a bioprotection context could be interesting to investigate. But the differences observed in the bioprotective effect of Mp2 and Mf1 strains and the lack of significant common biomarkers between these two conditions could suggest that interactions between Mp2 or Mf1 and the *Hanseniaspora* possibly involved different mechanisms.

## 4. Conclusions

Of the three bioprotective *Metschnikowia* strains tested in our experimental conditions simulating pre-fermentation phases at low temperatures, only two strains belonging to the species *pulcherrima* and *fructicola* showed an inhibitory effect on the growth of *Hanseniaspora* yeasts, especially in co-cultures with the lowest initial concentration of apiculate yeasts. This bioprotective effect was strongly reduced when the initial concentration of *Hanseniaspora* was of the same order of magnitude as the initial concentration of *Metschnikowia*, underlining the importance of quantifying the initial native yeast load of must to ensure the effective role of the added bioprotective strain.

The different inhibition kinetics observed between the two effective strains suggest different mechanisms of negative interaction. The *M. fructicola* and *M. pulcherrima* strains showed comparable consumptions of nitrogen resources. Amino acids such as lysine, leucine, isoleucine, methionine, and tryptophan were consumed preferentially by *Metschnikowia* strains and also by *Hanseniaspora*, suggesting that competition could occur for certain nitrogen resources. However, nitrogen consumption in all conditions (single or co-cultures) remained partial, indicating that nitrogen competition was probably not the mechanism involved in the bioprotective effect. Furthermore, oxygen was rapidly consumed in all conditions, but a bioprotective effect was observed only in some of them. These results seem to indicate that competition for oxygen was probably not involved in the effects observed. Although they offered interesting avenues for future investigations, particularly in relation to access to vitamin resources and peptide production, the untargeted metabolomic analyses did not clearly determine the metabolomic pathways involved or impacted by the interaction between bioprotective yeasts and *Hanseniaspora*.

The specific production of antimicrobial molecules remains to be investigated. Indeed, few bibliographical references mention the possibility of the synthesis of killer toxin-type molecules by *Metschnikowia* [50,51,52]. The production of this type of molecule under oenological conditions should be investigated more thoroughly.

## Figures and Tables

**Figure 1 foods-12-03927-f001:**
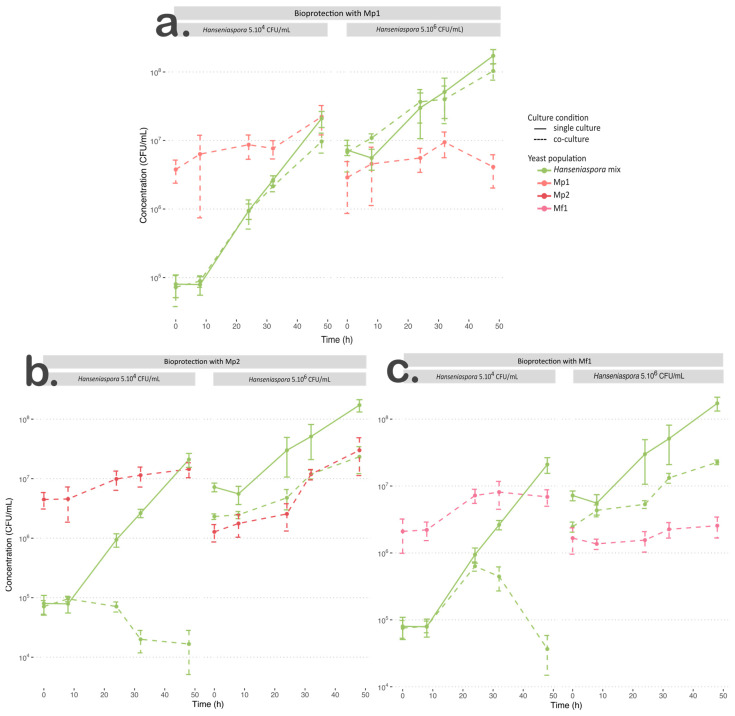
Growth curve of culture at 12 °C for *Hanseniaspora* mixture in single culture (green solid line) and co-culture (dotted line) with (**a**) Mp1, (**b**) Mp2, or (**c**) Mf1 at an initial *Hanseniaspora* concentration of 5.10^4^ CFU/mL and 5.10^6^ CFU/mL.

**Figure 2 foods-12-03927-f002:**
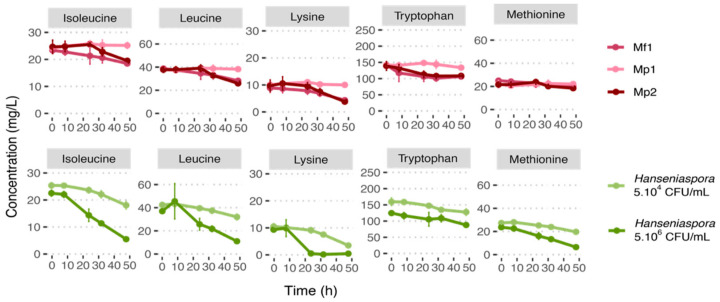
Nitrogen consumption curve at 12 °C in single culture during 48 h growth in synthetic must (MS300).

**Figure 3 foods-12-03927-f003:**
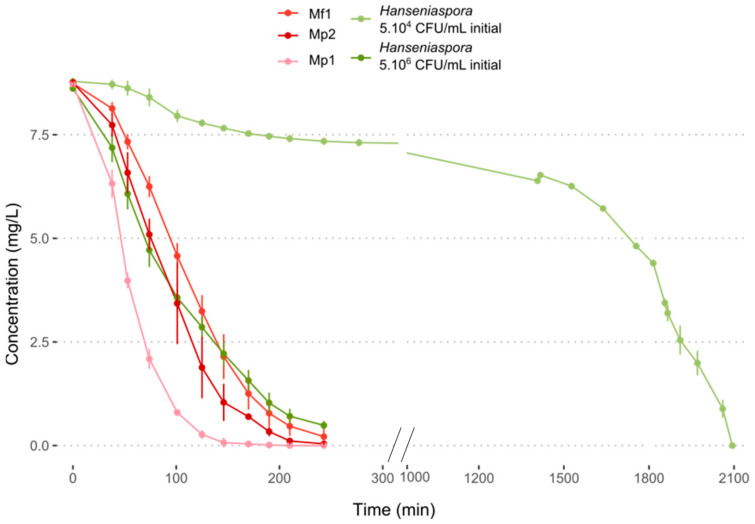
Oxygen consumption in synthetic must (MS300) at 12 °C.

**Figure 4 foods-12-03927-f004:**
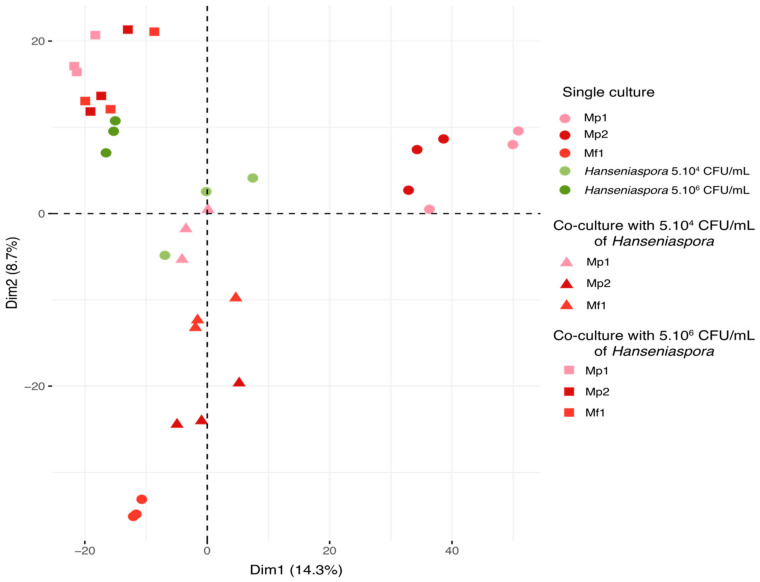
PCA of metabolic repartition for all samples in single and co-culture at 12 °C (3028 features).

**Figure 5 foods-12-03927-f005:**
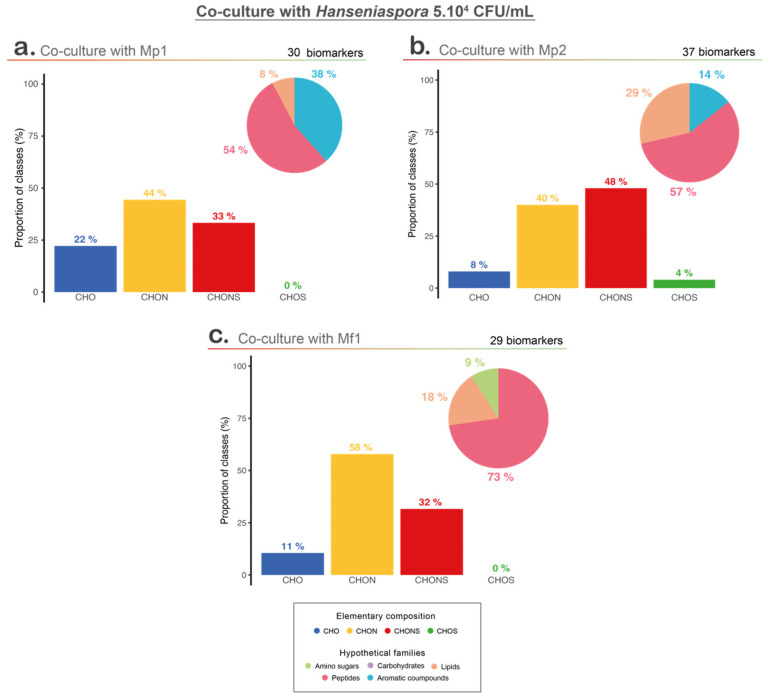
Metabolomic analyses with hypothetical families; Van Krevelen diagram and elementary composition of co-culture with 5.10^4^ CFU/mL of *Hanseniaspora* and (**a**) *M. pulcherrima* Mp1 strain, (**b**) *M*. *pulcherrima* Mp2 strain, and (**c**) *M. fructicola* Mf1 strain.

**Table 1 foods-12-03927-t001:** Initial concentration (expressed in CFU/mL) used for single cultures and co-cultures of Metschnikowia and Hanseniaspora mix.

	*Hanseniaspora* Mix	*Metschnikowia* (Mp1 or Mp2 or Mf1)
Single culture of *Hanseniaspora* mix	5.10^4^	-
	5.10^6^	-
*Metschnikowia* single culture	-	1.10^6^
Co-culture	5.10^4^	1.10^6^
	5.10^6^	1.10^6^

**Table 2 foods-12-03927-t002:** Growth parameters for *Metschnikowia* strains in single culture.

	Strain	Growth Factor ^1,2^	µmax (h^−1^)
Single culture	Mp1	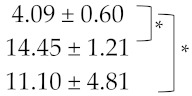	0.063 ± 0.020
	Mp2		0.062 ± 0.011
	Mf1		0.072 ± 0.006

^1^ Growth factor: Factor by which the population was multiplied between 0 h and 48 h of growth. ^2^ Brackets represent pair-wise comparison with the Kruskal–Wallis test; “*” corresponds to a statistical difference (*p* < 0.05).

**Table 3 foods-12-03927-t003:** Growth parameters of *Metschnikowia* strain in co-culture with *Hanseniaspora* mixture.

		Strains	*Metschnikowia* Final Population (CFU/mL)	*Metschnikowia* µmax (h^−1^)
*Metschnikowia* in co-culture with	*Hanseniaspora* at 5.10^4^ CFU/mL	Mp1	2.20.10^7^ ± 0.10.10^7^	0.063 ± 0.028
		Mp2	1.40.10^7^ ± 0.41.10^7^ *	0.039 ± 0.004 *
		Mf1	6.90.10^6^ ± 0.69.10^6^ *	0.074 ± 0.004
	*Hanseniaspora* at 5.10^6^ CFU/mL	Mp1	4.10.10^6^ ± 1.30.10^6^ *	0.029 ± 0.006 *
		Mp2	3.00.10^7^ ± 0.67.10^7^	0.197 ± 0.044 *
		Mf1	2.60.10^6^ ± 0.51.10^6^ *	0.056 ± 0.019

* Corresponds to values statistically different from values obtained in single culture (Kruskal–Wallis, *p* < 0.05).

**Table 4 foods-12-03927-t004:** Growth parameters of *Hanseniaspora* mix in single culture and co-culture with each *Metschnikowia* strain at 12 °C on synthetic must (MS300).

			Final *Hanseniaspora* Population (CFU/mL)	µmax (h^−1^)
5.10^4^ CFU/mL	*Hanseniaspora* single culture		2.10.10^7^ ± 0.25.10^7 a,a,a 1^	0.148 ± 0.014 ^a,a,a^
	*Hanseniaspora*	with Mp1	9.70.10^6^ ± 1.30.10^6 b^	0.134 ± 0.010 ^a^
		with Mp2	1.30.10^4^ ± 0.58.10^4 b^	0 ^b^
		with Mf1	3.70.10^4^ ± 2.00.10^4 b^	0.130 ± 0.012 ^a^
5.10^6^ CFU/mL	*Hanseniaspora* single culture		1.70.10^8^ ± 0.12.10^8 A,A,A^	0.094 ± 0.011 ^A,A,A^
	*Hanseniaspora*	with Mp1	1.00.10^8^ ± 0.23.10^8 B^	0.072 ± 0.015 ^A^
		with Mp2	2.30.10^7^ ± 0.58.10^7 B^	0.113 ± 0.027 ^A^
		with Mf1	2.30.10^7^ ± 0.15.10^7 B^	0.113 ± 0.008 ^B^

^1^ Letter corresponds to statistical groups (Kruskall–Wallis test, α = 5%), (in lower case letter) obtained by the pairwise comparison of values between single culture and co-culture at 5.10^4^ CFU/mL of initial *Hanseniaspora* mix, and (in capital letter) obtained by the comparison of values between single culture and co-culture at 5.10^6^ CFU/mL of initial *Hanseniaspora* mix.

**Table 5 foods-12-03927-t005:** Nitrogen consumption percentage for each strain at 12 °C after 48 h growth in synthetic must. Red bars represent all the resources consumed at more than 10% for Mp2, Mf1, and both initial concentrations of *Hanseniaspora* mix and dark grey bars those which were consumed less than 10%.

Nitrogen	Mp1	Mp2	Mf1	Hans_C4 ^1^	Hans_C6 ^2^
Aspartic acid	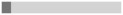 9 ± 10%	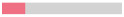 20 ± 34%	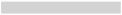 1 ± 2%	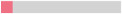 11 ± 7%	 63 ± 24%
Glutamic acid	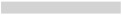 0 ± 0%	 3 ± 2%	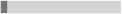 6 ± 5%	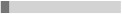 7 ± 11%	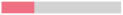 28 ± 4%
Alanine	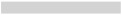 1 ± 1%	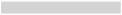 0 ± 0%	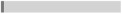 4 ± 6%	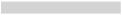 0 ± 0%	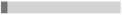 7 ± 8%
Ammonium	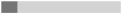 14 ± 5%	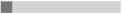 10 ± 9%	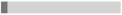 7 ± 6%	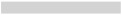 1 ± 1%	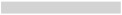 0 ± 0%
Arginine	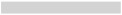 1 ± 1%	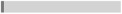 3 ± 4%	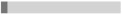 7 ± 7%	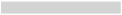 0 ± 0%	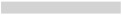 0 ± 1%
Cystine	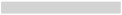 1 ± 2%	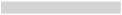 0 ± 0%	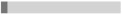 6 ± 6%	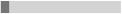 8 ± 9%	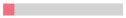 10 ± 9%
Glutamine	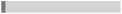 5 ± 5%	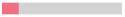 15 ± 13%	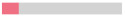 15 ± 13%	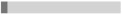 7 ± 6%	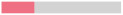 28 ± 5%
Glycine	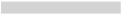 1 ± 1%	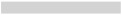 1 ± 1%	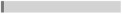 4 ± 3%	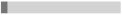 6 ± 10%	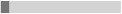 8 ± 4%
Histidine	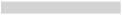 1 ± 1%	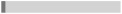 4 ± 8%	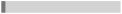 14 ± 7%	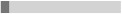 7 ± 12%	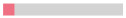 12 ± 4%
Isoleucine	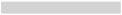 0 ± 1%	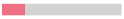 20 ± 7%	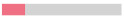 21 ± 3%	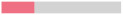 29 ± 8%	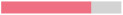 76 ± 2%
Leucine	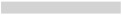 1 ± 2%	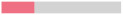 31 ± 4%	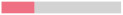 27 ± 5%	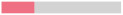 24 ± 9%	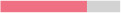 70 ± 2%
Lysine	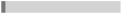 5 ± 9%	 60 ± 4%	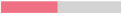 51 ± 6%	 66 ± 9%	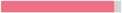 95 ± 4%
Methionine	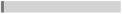 3 ± 6%	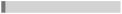 14 ± 10%	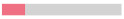 17 ± 5%	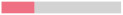 28 ± 10%	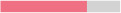 73 ± 2%
Phenylalanine	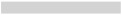 0 ± 0%	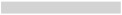 1 ± 1%	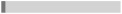 5 ± 4%	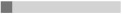 10 ± 12%	 35 ± 4%
Proline	 2 ± 4%	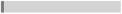 4 ± 5%	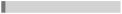 5 ± 8%	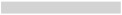 1 ± 2%	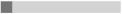 10 ± 15%
Serine	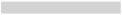 0 ± 0%	 2 ± 2%	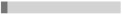 6 ± 5%	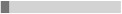 8 ± 11%	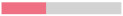 38 ± 3%
Threonine	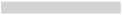 0 ± 0%	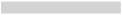 1 ± 1%	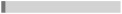 5 ± 4%	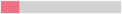 15 ± 10%	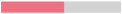 53 ± 2%
Tryptophan	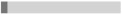 6 ± 4%	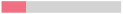 22 ± 7%	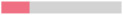 25 ± 2%	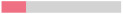 19 ± 11%	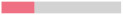 29 ± 4%
Tyrosine	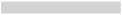 0 ± 0%	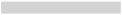 0 ± 0%	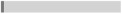 3 ± 3%	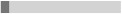 7 ± 13%	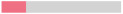 21 ± 4%
Valine	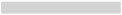 0 ± 0%	 2 ± 2%	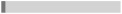 5 ± 4%	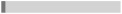 15 ± 7%	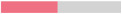 47 ± 3%

^1^ Hans_C4: Single culture of *Hanseniaspora* at initial concentration of 5.10^4^ CFU/mL. ^2^ Hans_C6: Single culture of *Hanseniaspora* at initial concentration of 5.10^6^ CFU/mL. Values are the mean ± standard deviation.

## Data Availability

The data used to support the findings of this study can be made available by the corresponding author upon request.

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
