# Peer review of "Bioprotection Efficiency of *Metschnikowia* Strains in Synthetic Must: Comparative Study and Metabolomic Investigation of the Mechanisms Involved"

_foods, 2023, doi:10.3390/foods12213927_

Round 1
Reviewer 1 Report
This study was aimed to investigate the efficiency of three Metschnikowia strains (two M. pulcherrima strains and one M. fructicola strain) to be used as bioprotective strains in a synthetic must, artificially contaminated with a mix of Hanseniaspora uvarum and H. valbyensis. The two Hanseniaspora strains were inoculated at two different levels in order to simulate two levels of contamination.
The paper is worthy of investigation, the research topic is highly innovative and of interest for scientific and winemaker communities. The materials and the experimental procedures are widely described. The results are properly presented and submitted to statistical analyses. The significance of results obtained was interpreted and discussed with reference to previously published studies.
Only few points have to be clarified.
Material and methods section
- Line 137: the authors report that “Populations of Hanseniaspora were counted on modified ITV selective medium….” indicating a bibliographic reference (Puyo et al., 2023), in which the same medium was used for the enumeration of Brettanomyces bruxellensis. Can the authors clarify this aspect? Why they did not use the same medium used for enumeration of Metschnikowia strains, the WL medium which is also useful for discrimination of Hanseniaspora strains?
- An editorial problem occurs for Table S1 as the values reported in the first column are overlapped with line numbers.
Result section
- Table 2, page 6. I suggest to the authors to give more details regarding the growth parameters reported in this table, i.e. the method used for calculation of the Growth factor.
- I suggest to check the results obtained with PCA (reported in the figure 4) as the percentage of variance explained along the two dimensions is quite low (14.3% for dimension 1 and 8.7% for dimension 2).
Reviewer 2 Report
Review of Manuscript entitled “Bioprotection efficiency of Metschnikowia strains in synthetic must: comparative study and metabolomic investigation of the 3 mechanisms involved” for Foods (Foods- 2647085)
Manuscript is well-designed and the topic is original. Introduction part includes appropriate information. MM section needs some revisions. Results and discussion section is well design and includes enough content. Also, I believe that descriptive expression will increase the effectiveness of the manuscript and avoid the complication for R&D seciton. I strongly recommend to give the explanations and the long version of the abbreviations. For the conclusion section, please do not prefer sentences that includes references. Additionally, figures and tables should be improved for providing proper visuality of the manuscript.
I recommend a minor revision.
These advises could be taken in consider for improving manuscript:
Page 2, Line 73: Three commercial yeasts provided from the French Wine and Vine Institute (Institut Francais de la Vigne et du Vin-IFV, Nantes, France).
Page 2, Line 82: manufacturer's (IFV) instructions
Page 2, Line 89: μL
Page 2, Line 91: μL/min
Page 2, Line 92: Please check and revise ‘μL ‘ as ‘μL’.
Page 3, Line 130: Wallerstein Laboratory (WL) CM0309 agar medium (Oxoid, Hampshire, UK)
Page 3, Lines 142-143: Please provide more information about “Primary Amino Nitrogen (PAN)” and “Ammonia” kit (BioSystem, Muttenz, Switzer- 142 land).
Page 4, Line 147: Please delete the Table S1. It can be explained as a sentence in MM.
Page 4, Line 159: Please insert reference for the methodology
Page 4, Line 165: 2.6 μm
Page 4, Lines 168-172: Please rephrase the sentence
Page 4, Lines 168-172: (45/45/10, v/v/v)
Page 4, Lines 175 and 177: Please insert more information about OPA and FMOC.
Page 4, Line 184: Equipment name (Model name, brand name, city name, country name) for MS300
Page 5, Line 198: Please insert ‘10500xg’ as rpm and Equipment name (Model name, brand name, city name, country name) for the centrifuge
Page 5, Lines 215-216: Please insert more information about ‘Smart Formula’
Page 5, Line 216: Software
Page 5, Line 216: Please revise ‘20 and 5 ppm’ as ‘5 and 20 ppm’
Page 5, Line 219: Please correct as ‘Lipids’ and correct all manuscript in terms of similar corrections.
Page 5, Line 221: Please correct ‘Rivas-Ubach et al. (2018) [21]’ as ‘Rivas-Ubach et al. [21]’
Page 5, Line 221: Please correct ‘Schymanski et al. (2014) [22]’ as ‘Schymanski et al. [22]’
Page 5, Line 221: Please explain and give more information about: YMDB and KEGG.
Page 5, Line 227: Please provide more information about RStudio Software (Version 4.2.2, City name, Country name)
Page 5, Line 237: 32 h instead of 32 hours (Please, make similar corrections for all manuscript)
Page 6, Line 247: Please insert extra space after ‘±’ (4.09 ± 0.60) and make similar corrections for all manuscript.
Page 6, Line 256: Please revise all ‘(p-value < 0.05)’ as ‘(p<0.05)’ for all manuscript.
Page 7, Line 306: 2.37 ± 0.23 107 CFU/mL and please provide similar corrections for all manuscript.
Page 9, Line 402: Please correct ‘Su et al. (2020) [27]’ as ‘Su et al. [27]’. Please check and provide similar corrections for all manuscript.
Page 9, Line 402: Please delete ‘who also’
Page 9, Line 421: 8 h
Page 10: Table 5. HansC4, HansC6
Page 10: Please explanin the values that are given in Table 5. Are they given as mean ± SD? If yes, please insert information as foot-note. Also, SD or SEE or some given values are higher than the mean value. 20% ±34%, 7% ±11%? Please check and revise values given in the Table 5. And revise values as mean ± SD % or mean ± SEE %.
Page 10, Line 253: ng/L/h/cell ?
Page 11, Line 461: Please explain and give more information about: YAN
Page 11, Line 502: UHPLC-Q-TOF-MS/MS
Figure 1: Please revise coculture as co-culture
Please check reference list, some journal names are given in abbreviation.
Page 13, Line 568: Please insert more information about CHON.
Page 14, Line 663: B6
Page 17, Line 686: Please avoid using references for conclusion.
For all figures, please use black color and Palatino Linotype font.
English usage is well.
Minor revision could increase the effectiveness of the manuscript.
Reviewer 3 Report
This study highlights the strain-specific dynamics and metabolic footprint of Metschnikowia bio-protection yeasts in winemaking, offering valuable insights for optimizing wine fermentation processes.
The abstract effectively highlights the critical role of strain selection and the importance of the initial ratio between Metschnikowia and Hanseniaspora in ensuring successful bioprotection in winemaking.
In introduction section, It would be beneficial to elaborate on the "negative interactions" mentioned between bioprotective strains and indigenous flora. What specific negative interactions are being investigated or hypothesized? With references
It's interesting that metabolomic footprint analysis is mentioned as part of the study. Providing more information on the specific metabolites or pathways being analyzed and their relevance to bio-protection would enhance the understanding of the research.
The text mentions two levels of contamination, but it could be unclear why these specific levels were chosen and how they relate to real-world winemaking conditions.
Given the complexity of the experimental design, consider including a flowchart or diagram to visually represent the different culture conditions, sampling points, and analyses. This can aid in understanding the study's structure.
The inclusion of Table S1 for the eluant gradient used in amino acid analysis is helpful. Ensure that this table is properly referenced within the text for easy access.
Ensure consistency in terminology throughout the section. For example, the use of "Metschnikowia mix" and "Metschnikowia single culture" could be made more consistent to avoid any potential confusion.
The conclusion appropriately acknowledges the complexity of the interactions between Metschnikowia and Hanseniaspora yeasts during pre-fermentation phases. Consider conducting focused research to elucidate the specific antimicrobial molecules produced by Metschnikowia strains for a more in-depth understanding of their bioprotective mechanisms.
It's crucial to ensure that all references and citations are correctly included once the paper is finalized.
Moderate editing of English language required
Round 2
Reviewer 2 Report
Manuscript has improved well.
But still for values given in Figure 5, SD values are so high.
Please reconsider this.
